# Discriminant Canonical Analysis as a Validation Tool for Multivariety Native Breed Egg Commercial Quality Classification

**DOI:** 10.3390/foods10030632

**Published:** 2021-03-17

**Authors:** Antonio González Ariza, Ander Arando Arbulu, Francisco Javier Navas González, Juan Vicente Delgado Bermejo, María Esperanza Camacho Vallejo

**Affiliations:** 1Department of Genetics, Faculty of Veterinary Sciences, University of Córdoba, 14071 Córdoba, Spain; angoarvet@outlook.es (A.G.A.); anderarando@hotmail.com (A.A.A.); juanviagr218@gmail.com (J.V.D.B.); 2Instituto de Investigación y Formación Agraria y Pesquera (IFAPA), Alameda del Obispo, 14004 Córdoba, Spain; mariae.camacho@juntadeandalucia.es

**Keywords:** egg quality, external quality traits, internal quality traits, DSM color, fan color coordinate decomposition, mechanical eggshell strength, pH-related traits

## Abstract

This study aimed to develop a tool to validate multivariety breed egg quality classification depending on quality-related internal and external traits using a discriminant canonical analysis approach. A flock of 60 Utrerana hens (Franciscan, White, Black, and Partridge) and a control group of 10 Leghorn hens were placed in individual cages to follow the traceability of the eggs and perform an individual internal and external quality assessment. Egg groups were determined depending on their commercial size (S, M, L, and XL), laying hen breed, and variety. Egg weight, major diameter, minor diameter, shell b*, albumen height, and the presence or absence of visual defects in yolk and/or albumen showed multicollinearity problems (variance inflation factor (VIF) > 5) and were discarded. Albumen weight, eggshell weight, and yolk weight were the most responsible traits for the differences among egg quality categories (Wilks’ lambda: 0.335, 0.539, and 0.566 for albumen weight, eggshell weight, and yolk weight, respectively). The combination of traits in the first two dimensions explained 55.02% and 20.62% variability among groups, respectively. Shared properties between Partridge and Franciscan varieties may stem from their eggs presenting heavier yolks and slightly lower weights, while White Utrerana and Leghorn hens’ similarities may be ascribed to hybridization reminiscences.

## 1. Introduction

In 2019, the world’s hen egg production exceeded 1.6 billion eggs, 28.7% higher production than a decade before [1]. Such a remarkable increase brought about a parallel increase in the concerns for animal welfare and environment in the European Union. Contextually, more than 50% of hens were reared in cage-free systems, while 18% of hens were reared in alternative production systems (free-range and organic) in Europe in 2019 [2].

This increasing interest in products obtained under non-industrial production systems allows the development of sustainable farming practices [3]. These sustainable farming practices may involve the use of native breeds adapted to the local environment, with great rusticity and resistance to meteorological situations and diseases, as well as great ability to search for food in the wild [4,5]. Consequently, it is through the conservation of animal genetic resources, that economic sustainability in the rural areas is promoted [6,7].

The Utrerana avian breed is one example of rustic Spanish hen, located in Andalusia (Southern Spain), which is officially considered to be endangered as stated in the Royal Decree—Law 45/2019 from the 8 February 2019. Its four varieties, namely, White, Franciscan, Black, and Partridge, are classified depending on the color of its feathers and tarsi [8]. It was initially oriented toward egg production, however, the introduction of rather productive commercial hybrid genotypes in Europe caused the displacement of the Utrerana hen breed to a secondary position [9]. As a result, the breed census reached a critical situation, which was parallel to a decrease in its productive indices derived from the patent lack of productive selection [10]. Although the number of individuals has multiplied in the last years, only 1548 animals were registered in the studbook of the breed during 2019 [11].

The enhancement of local products can be a strategy for the conservation of autochthonous genotypes, by avoiding the loss of connection between products, the local breeds from which they derive, and the area in which these were produced, as has been described for industrial products [12]. In line with this situation, the definition of the breed’s productive role became compulsory to maximize the breed’s potential to satisfy current commercial demands. The characterization of Utrerana’s egg as the main product of the breed was configured in the context of a set of strategies that sought the obtention of competitive sustainable products in the framework of the recent emerging diseases and climate change [13].

The acceptability of the eggs by consumers is mainly affected by the characteristics that describe their quality. Egg quality depends on several parameters, which are related to the eggshell, the albumen, and the yolk. Quality traits can be classified into external and internal quality, depending on whether the egg has to be broken to be scored (internal quality) or not (external quality) [14,15]. Egg weight, eggshell strength, albumen quality, and yolk color intensity are among the most important egg traits of commercial interest [14,16,17,18]. Eggs are commercially classified into four classes depending on their total egg weight: S (<53 g), M (53–63 g), L (63–73 g), and XL (>73 g).

Egg quality traits have been reported to be multifactorially dependent mainly on the laying hen’s age and nutritional factors [16,19,20,21]. However, there are some relevant pieces of evidence for the influence of the genotype on some of these egg quality traits on the relative proportion of yolk and albumen, albumen quality, or chemical composition [14,22,23,24,25,26].

Utrerana’s egg not only constitutes a differentiated product in terms of internal and external quality–related traits [14] and chemical composition [23]. Additionally, its sensory characteristics have been reported to differ from the eggs of commercial lines [27].

Consequently, the present study aimed to determine the contributions of external and internal quality parameters to the eggs produced by each of the four varieties of Utrerana hens and a control flock of a commercial laying lineage. Canonical discriminant analysis was used to design a statistical tool that permits determining whether specific eggs may correctly fit the features of the different commercial size categories (S, M, L, and XL), which may support the standardization of the Utrerana varieties’ eggs as products and may address and support their suitability to cover particular sections of the market for egg consumption. 

## 2. Material and Methods

### 2.1. Institutional Animal Care and Use Committee Statement

Avian-specific codes for good practices and the national guidelines for the care and use of laboratory and farm animals were followed in agreement with the standards found under the scope of the European Union legislation (2010/63/EU, from the 22 September 2010) and its transposed Spanish law document (Royal Decree Law 53/2013). As recommended by Royal Decree Law 53/2013 and its credited entity, the Ethics Committee of Animal Experimentation from the University of Córdoba, no additional permission was required as stated in the 5th section of the 2nd article of the aforementioned document given the zootechnical credited utilization of the animals participating in the present study.

### 2.2. Layer Flock and Environmental Conditions

The public farm in which the study took place is located at the Agropecuary Provincial Center of Diputación of Córdoba, south Spain (Plus code: W77Q + MF El Levigar/37°54′50.9″ N 4°42′40.4″ W). The eggs used in the experiment were obtained from a layer flock comprising 60 Utrerana hens and 10 Leghorn Lohmann LSL-Classic lineage hens (hereinafter referred to as “Leghorn hens”), distributed depending on their breed or variety as described in Table 1. The laying flock was housed in individual cages (50 × 62 × 41 cm), to ensure that the traceability of each egg daily was feasible and following the Council Directive 1999/74/EC of 19 July 1999, which states minimum standards for the protection of laying hens. This whole year study ran from February 2019 to February 2020. All the birds were fed on the same commercial feed (15.20% crude protein, 4.60% crude fat and oils, 3.20% crude fiber, 14.00% crude ashes, 4.10% calcium, 0.66% phosphorus, 0.19% sodium, 0.31% methionine, 0.72% lysine). Feed and water were available *ad libitum*.

### 2.3. Work Sample

All statistical tests were performed on an egg sample comprising 541 eggs, laid during a complete laying cycle. The eggs were classified depending on their breed and variety and commercial size as shown in Table 2. The protocols are described below in Section 2.4 and Section 2.5 were performed on each egg individually.

### 2.4. External and Internal Quality-Related Traits Description

External and internal quality-related traits were measured separately. For the external quality of the egg, noninvasive methods were used, and measurements were taken without breaking the eggshell. The external quality traits that were measured were as follows: egg weight; major and minor diameters of the egg; eggshell color lightness, redness, and yellowness coordinates (shell L*, shell a*, and shell b*). Shape index (SI) was computed through the following formula [28]:SI = (ØM/Øm) ∗ 100(1)
where ØM is the major diameter and Øm is the minor diameter.

Eggs were classified depending on their shape index as follows: sharp egg (SI < 72), standard egg (SI = 72–76), or round egg (SI > 76) [29]. 

Internal egg quality–related traits were evaluated after breaking the egg. Internal egg quality traits measures were as follows: eggshell resistance (eggshell strength and area under the force–displacement curve); albumen height; yolk color; yolk lightness, redness, and yellowness variables (yolk L*, yolk a*, and yolk b*); yolk diameter; eggshell weight; yolk weight, albumen weight; yolk pH; albumen pH; eggshell thickness; and the presence or absence of visual defects in yolk and/or albumen. Haugh units (HU) were calculated as a measure of the albumen quality, from the variables albumen height and egg weight via the following formula [30]:(2)$$\mathrm{HU}=100*\log\left(h-1.7w^{0.37}+7.6\right)$$
where h is albumen height (mm) and w is egg weight (g).

### 2.5. Measurements on Eggs

The egg quality measurements were registered fortnightly for the whole duration of the study. Egg quality was assessed at 22 ± 1 °C. The traceability of the egg and the external and internal characterization of each individual egg were performed and registered within 24 h after oviposition. Eggs were weighed individually using an electronic scale (Cobos, CSB-600C, Barcelona, Spain). Eggshell color was assessed using a portable spectrophotometer (CM 700d, Konica Minolta Holdings Inc., Tokyo, Japan). Eggshell color results were expressed using the International Commission on Illumination (CIE) L*a*b* system color profile. Major and minor diameters were measured using a Vernier scale (Electro DH M 60.205, Barcelona, Spain).

Mechanical eggshell strength measurement was performed using a texturometer TA.XT2 Texture Analyzer (TA.XT2; Texture Technologies Corp., Scarsdale, NY, USA). Eggshells were punctured at the bottom (large end) of the eggshell with a polyoxymethylene (POM) probe with a 5 mm diameter. Eggshell strength and the area under the curve were determined from the graphical curve obtained by the texturometer. The approach followed started when each individual eggshell was broken, and the yolk and albumen were deposited on a glass surface to take measurements of the internal quality–related variables described above. Albumen height was computed as the arithmetic mean of three measurements performed using a Haugh digital micrometer (Baxlo, Barcelona, Spain). The intensity of the yellow–orange color of the yolk was measured both with the portable spectrophotometer (L*, a*, b*) and using a Roche color fan (yolk color) (DSM, DSM^®^ YolkFan^TM^, Heerlen, The Netherlands). The yolk diameter was measured on a Vernier scale. The eggshell, albumen, and yolk were separated and weighed using a precision balance. The pH was measured using a pH meter (Crison^®^, PH-25, Barcelona, Spain). Eggshell thickness was measured averaging three measurements around the blow-hole near the equator of the egg upon a Vernier scale. All the eggs were visually evaluated to detect blood or meat spots in the albumen, yolk, or both.

### 2.6. Canonical Discriminant Analysis

A canonical discriminant analysis was performed using egg weight, major diameter, minor diameter, shell L*, shell a*, shell b*, shape index, eggshell strength, area under the force–displacement curve, albumen height, Haugh units, yolk color, yolk L*, yolk a*, yolk b*, yolk diameter, eggshell weight, yolk weight, albumen weight, yolk pH, albumen pH, eggshell thickness, presence or absence of visual defects in yolk and/or albumen, and Haugh units per egg as explanatory variables. The commercial classification and the hen breed/variety were used as the labeling classification criteria to measure the variability in quality-related traits between and within classification groups, to establish, identify, and outline clusters [31,32].

The present discriminant tool permits to sort eggs across hen genotype and quality categories and to determine the clustering patterns described by the egg sample through a linear combination of quality-related traits. Canonical discriminant analysis was also used to plot pairs of canonical variables building a territorial map to graphically interpret group differences. Variable selection was performed using regularized forward stepwise multinomial logistic regression algorithms as suggested by Marín Navas et al. [32]. Priors were regularized based on group sizes computed from the prior probability option in SPSS version 26.0 software rather than considering them to be equal, to prevent group with different sample sizes from affecting the quality of classification [33]. As the previous authors suggested, the statistical analysis used in the present research has been reported to be robust when sample sizes between groups are highly unequal. To palliate potential distortion effects, the smallest sample size should be at least 20 for every 4 or 5 predictors, and the maximum number of independent variables should be n − 2, where n is the sample size [34]. However, the fact of having 4 or 5 times more observations and dependent variables than previously described makes the discriminant approaches efficient [32]. This requirement is far surpassed in the present study, so the distorting effects mentioned are avoided.

Multicollinearity is a statistical phenomenon in which two or more variables are reciprocally dependent upon other variables in a way such that one can be linearly predicted from the rest with a high degree of accuracy. Multicollinearity analysis was performed before discriminant analysis to ensure that the regressors used were independent, so the variables chosen by the forward or backward stepwise selection methods were the same. Then, the forward stepwise selection method was chosen, as it is less time-demanding than the backward selection method.

Canonical discriminant analysis was performed by the use of the Discriminant routine of the Classify package of the SPSS version 26.0 software and the Discriminant Analysis routine of the Analyzing Data package of XLSTAT Pearson Edition.

#### 2.6.1. Multicollinearity Preliminary Testing

The multicollinearity assumption was tested to discard redundancies in the variables considered so that this phenomenon does not condition the structure of the matrices or overinflate the explanatory potential of variance, before performing a discriminant canonical analysis [32]. The variance inflation factor (VIF) was computed and used as an indicator of multicollinearity, following the formula:(3)$$\mathrm{VIF}=1/\left(1-R^{2}\right)$$
where R^2^ is the coefficient of determination of the regression equation. A VIF value of 5 was accepted in the present research, as reported by other authors [35]. The amount of variability in a certain independent variable that is not explained by the rest is called the tolerance and is calculated as 1 − R^2^ [36]. If tolerance has values lower than 0 and, simultaneously, the value of VIF is ≥10, multicollinearity can be considered a problem. For this, the Linear routine of the Regression package of the SPSS, version 26.0 software was used.

#### 2.6.2. Canonical Correlation Dimension Determination

The maximum number of canonical correlations (interpreted as Pearson’s ρ) between two sets of variables is the number of variables in the smaller set. Although the first canonical correlation may often explain most of the relationship between sets, all canonical correlations must be considered [37]. Canonical correlation values of ≥0.30 may be indicative of a statistically significant dimension.

#### 2.6.3. Canonical Discriminant Analysis Efficiency

Variables that may significantly contribute to the discriminant function are evaluated by Wilks’ lambda test. As Wilks’ lambda approximates to 0, the contribution of the variable to the discriminant function increases. Functions can be used to explain group ascription if the significance (tested using χ^2^) is below 0.05 [38].

#### 2.6.4. Canonical Discriminant Analysis Model Reliability

The assumption of equal covariance matrices was evaluated through Pillai’s trace criterion, which is the only acceptable test to be used in cases of unequal sample sizes [32,39]. Pillai’s trace criterion was computed using the Multivariate routine of General Linear Model package of the software SPSS, version 26.0 software. Statistical differences in the dependent variables across the levels of independent variables are considered when significance is below 0.05.

#### 2.6.5. Variable Dimensionality Reduction

The overall variables were minimized to a few significant variables that contributed most to the different variations in the different types of eggs using a preliminary principal component analysis (PCA).

#### 2.6.6. Canonical Coefficients and Loading Interpretation and Spatial Representation

The percentage of allocation of an egg within its group (defined by its commercial size and the genotype of the hen that laid it) was determined using a discriminant function analysis. The variables that presented a discriminant loading of ≥|0.40|, were considered to be substantially discriminant. Non-significant variables were excluded from the function using stepwise procedures. The larger the absolute coefficients for each particular variable within a set, the better the discriminating ability [32]. Data were standardized following the premises described by Manly and Alberto [40]. Afterward, squared Mahalanobis distances were calculated. Squared Mahalanobis distances between groups were obtained using the following formula:(4)$$D_{\mathrm{ij}}^{2}=({Ῡ}_{i}-{Ῡ}_{j}){\;\mathrm{COV}}^{-1}({Ῡ}_{i}-{Ῡ}_{j})$$
where $D_{\mathrm{ij}}^{2}$ is the distance between population i and j; ${Ῡ}_{i}$ and ${Ῡ}_{j}$ are the means of variable x in the ith and jth populations, respectively; COV^−1^ is the inverse of the covariance matrix of measured variable x.

The squared Mahalanobis distance was used to graphically depict the clustering patterns defined by the differences in the values for quality-related traits across the potential egg classifications considered in the present research. To this aim, a dendrogram representing the possible categories within egg quality classification was constructed using the underweighted pair-group method arithmetic averages (UPGMA) from the Universität Rovira i Virgili (URV), Tarragona, Spain, and the Phylogeny procedure of MEGA X 10.0.5 (Institute of Molecular Evolutionary Genetics, The Pennsylvania State University, State College, PA, USA).

#### 2.6.7. Discriminant Function Cross-Validation

The hit ratio can be defined as the percentage of correctly classified observations [41]. The leave-one-out cross-validation option was used to validate the discriminant functions used. The classification rate must be at least 25% higher than obtained by chance to be considered accurately enough [32].

Press’ Q significance test was used to compare the discriminating power of the cross-validated function by using the following formula:(5)$$\mathrm{Press}’Q={\left[N-\left(\mathrm{nK}\right)\right]}^{2}/\left[N\left(K-1\right)\right]$$
where N is the number of observations in the sample; n is the number of observations correctly classified; and K is the number of groups. Subsequently, the value of Press’ Q statistic was compared with the critical value of 6.63 for χ^2^ with one degree of freedom in a significance of 0.01. If Press’ Q exceeds the critical value of χ^2^ = 6.63, the cross-validated classification can be considered significantly better than chance.

## 3. Results

### 3.1. Canonical Discriminant Analysis Model Reliability

Egg weight, major diameter, minor diameter, shell b*, albumen height, and presence or absence of visual defects in yolk and/or albumen showed VIF values over 5 and were discarded from further analyses. A summary of the value of tolerance and VIF for each variable is shown in Table 3.

Pillai’s trace criterion reported a significant difference across the different egg quality classification groups considered in the study (*p* < 0.05; Table 4).

### 3.2. Canonical Coefficients, Loading Interpretation, and Spatial Representation

Six discriminating canonical functions were identified in the discriminating canonical analysis (Table 5). Table 6 reports the outcomes of discriminating ability testing. Higher eigenvalues were indicative of higher discriminatory power. Functions F1 and F2 with eigenvalues greater than 1 explain 75.63% of the total variance, while the rest contribute to the explanation of the variance with a low percentage of the information to the analysis.

After discarding redundant variables, the test of equality of group means across egg quality classification groups was used to rank variables depending on their discriminating properties (Table 7).

The greater the value of F and the lower the value of Wilks’ lambda for a certain variable, the better its discriminating power was, and hence, the higher its position in the rank was as well.

As shown in Table 8, standardized discriminant coefficients were evaluated. This allowed us to determine the possibility of a reduction in the discriminant power of individual variables as a result of multicollinearity between pairs.

The substitution of the values for quality-related traits in the first two discriminating functions was performed to obtain *x* and *y*-axis coordinates, for the first and second dimensions, respectively. Once coordinates were obtained, each egg observation was sorted and classified across the different egg quality classification categories and laying hen genotype. Coordinates were used to depict eggs on a territorial map (Figure 1). Centroids represent the means of the discriminant function scores by egg quality classification group for each function calculated.

In this regard, Mahalanobis distances were used as they represent the probability that a case of an unknown background belongs to a particular egg quality classification group. It can be calculated through the relative distance of the problem egg to the centroid of its closest group. The probability of classification of observation into a group was calculated, following the premises in Hair et al. [42].

Consequently, the hit ratio, or successfully classified cases, was determined (Appendix A). Mahalanobis distances obtained after the evaluation of the discriminant analysis matrix were transformed into squared Euclidean distance and represented in Figure 2.

### 3.3. Discriminant Function Cross-Validation

Classification and leave-one-out cross-validation matrices were evaluated (Appendix A). In all, 73.2% of original grouped cases were correctly classified in the different egg quality classification group, from which 57.1% of clustered observations were cross-validated. A Press’ Q value of 5297.17 (N: 541; n: 396; K: 20) was obtained; hence, predictions were considered to be significantly better than those that would be obtained by chance at 95% [43].

## 4. Discussion

Involving autochthonous breeds in animal production systems may promote the evolution of sustainable ways of producing. Native breeds can be used in search of productive improvement by taking advantage of genotypes adapted to the climatology and orography, as well as to the technical, productive, and cultural conditions of the area. On the other hand, commercial chains are increasingly requesting more products derived from non-industrial processes. This context makes it necessary to characterize the eggs of the Utrerana avian breed according to their commercial size while defining how the different quality-related parameters affect the differentiation between eggs across the different varieties and breeds studied. The results obtained in the present study may suggest how to approach the different strategies to make an endangered breed profitable, thus ensuring its conservation by establishing production models to which it is adapted.

The selection of the individuals in the sample was performed considering that the hybrid commercial cycle and both genotypes reach 50% of laying (egg production during a laying cycle). Contextually, the typical production cycle in commercial layers (Leghorn hens among others) lasts about 72 weeks [44]. However, this cycle may extend until 156 weeks in around a third of the Utrerana population [45]. Additionally, according to Kuo et al. [46], the age at sexual maturity is estimated by age in weeks when 50% egg production is reached. In this regard, the same authors suggested the age when 50% egg production in White Leghorn is reached to be around 21 weeks. By contrast, the information reported by Orozco Piñán [45] suggested the average age of Utrerana hens at the moment of the first laying was 25 weeks. Furthermore, the breeding criterion of both breeds may differ, as while White Leghorn hens breeders have traditionally selected animals for precocity [47], Utrerana breeders have not sought this trait as a priority rather benefiting from the natural lay cycle of the breed [45]. Zita et al. [48] suggested that egg quality characteristics are affected by the interaction of genotype (breed and strain) and hen’s age, rather than exerting their effects independently.

Multicollinearity analyses revealed high correlations between major diameter and minor diameter and egg shape index, since both measurements comprise the formula for its calculation. The same happens with the formula of Haugh units, which includes the variables of egg weight and albumen height, which consequently were eliminated due to multicollinearity problems. Moreover, egg weight can be calculated by separately summing albumen weight, yolk weight, and eggshell weight variables, which may be the logical source for the redundancies detected.

Degree of lightness (L*) and chromaticity coordinates (a* and b*) comprise the L*a*b* color space [49]. In this context, coordinates of shell a* and shell b* are difficult to interpret and can be correlated in white-shelled eggs, such as those of the Utrerana avian breed [20]. As suggested by other authors [50], a* and b* parameters measure chromaticity. More specifically, redness–greenness and yellowness–blueness, respectively. Positive values of a* are linked to increased amounts of redness in eggshell color, whereas negative values of a* relate to increased amounts of greenness in the eggshell color.

Similarly, the representativity of yellow and blue components in eggs of any color are represented by positive and negative values of b*, respectively. In this context, Odabaşi et al. [50], suggested that the lighter the shell color (higher L*), the lesser the redness of the color of the eggshell is as well. This was in line with the results reported by Aygun [51], who reported shell L* could be considered as a discriminative color criterion as the lesser the amount of shell L*, the darker the eggshell color turns to be.

The visual defects in yolk and albumen are produced by meat and blood spots. The presence of these visual defects is regarded as an undesirable feature in eggs that causes rejection by consumers [52].

These undesirable findings may stem from the synthesis of the different parts of the egg during ovulation due to the rupture of an ovarian follicle at a different position from the stigma [53]. In these situations, variations in the chromaticity coordinates of the yolk color could appear, thus, may be one of the sources of multicollinearity problems between the presence of visual defects and yolk a* and yolk b*.

Albumen weight, eggshell weight, and yolk weight variables reported the best discriminating properties (Table 7). These three quality-related traits compose the egg weight, which is the main criterion on which the commercial classification of eggs relies. At the same time, albumen represents about 55–65% of the egg weight [54,55]. This explains the fact that albumen weight was ranked first at the test of equality of group means.

Hen strain has been reported to significantly affect albumen ratio [53,56]. Albumen is critical for the survival of the chicken embryo and the variations in the content of albumen in hen eggs can generate differences in skeletal muscle or liver metabolism during embryonic development [57]. In laying hens, albumen has great commercial importance, provided its unique functional properties and its use as an ingredient in a large number of culinary international preparations [58]. In previous studies, the Leghorn has been demonstrated to have a higher albumen weight than the Utrerana avian breed, due to the Leghorn’s higher concentration of energy reserves [14]. Contextually, Peña-Villalobos et al. [57] suggested a significant reduction in metabolic rate occurs in the last fifth of embryonic life in albumen-removed eggs, which in turn derives into reduced catabolic activities in the skeletal muscle of chicks that eventually hatch.

Utrerana has been reported to present a lower eggshell weight and a higher yolk weight than Leghorns [14]. These results agree with the present research since these parameters have a high discriminating power when clusters differentiate. Modern commercial breeds showed clear differences in terms of eggshell weight when compared to native poultry, due to the high selection of all egg traits of eggs for its transport and commercial purposes [59,60].

Differences in the proportion of egg yolk have been reported between breeds and within highly productive laying hens strains such as the White Leghorn, which may be indicative of the presence of sufficient additive genetic variation [61]. Furthermore, selection based on additive genetic variation in yolk weight has been suggested as an option to promote seeking sustainability of local eggs [62], as native breeds could satisfy the growing demand for more energetically efficient eggs in the market [13,14,26].

Yolk diameter and Haugh units reported the best discriminating properties (fourth and fifth position in the rank) after weight-related traits (albumen, eggshell, and yolk weights). The relevance of these traits may be ascribed as suggested by Ukwu et al. [63], who reported significant differences in yolk weight and albumen height among light (less than 49.99 g), medium (50–55 g), and heavy eggs (more than 55 g) of Isa Brown egg layer chickens in Nigeria. This has also been reported by Alkan et al. [64], who addressed a parallel increase in yolk diameter as egg weight increases in partridge eggs. However, no differences between Utrerana and Leghorn breeds were detected in previous studies [14].

Haugh units are used as an indicator of internal egg quality [30]. Time of storage and storage conditions affect Haugh units values [65]. However, the strains or breed of the hen have been reported to quantitatively affect them. For instance, several authors have reported higher values for Haugh units in local breeds than in commercial hybrid strains [4,13,66]. In any case, albumen height is correlated with the percentage of albumen [67]. Hence, commercial strains could present a certain advanced position, provided a larger percentage of albumen is found in hybrid strains in comparison to that in native breeds.

Values for pH-related traits showed the lowest values of F and highest for Wilks’ lambda. Egg pH allows the assessment of the egg’s freshness [68,69]. The loss of CO_2_ and H_2_O inside the egg produces an increase in albumen pH. The time of storage and high temperatures condition this loss of CO_2_ and H_2_O and promote a decrease in albumen viscosity and flavor with detrimental effects for egg quality [19,70]. Albumen and yolk pH can be slightly influenced by the hen strain [13,14,71]. Nevertheless, in the present study, when all egg pH values were measured during the 24 h following oviposition, it was found that albumen pH and yolk pH have a low discriminating power between different groups of eggs, which may derive from the low variability in pH found. Such lack of variability may stem from the fact that the eggs considered in this study were fresh enough for those eggs presenting slightly lower values not to be detrimental on egg quality. Additionally, this finding may evidence a patent lack of importance provided to quality traits (such as the pH of the components of the egg) against quantitative traits among the current criteria that are considered for egg quality classification, as the quantity of the product may be better commercially valued than its quality. However, this commercial strategy may be erroneous given it may not match the current general trend of the customers preferring egg quality over quantity [72].

Figure 2 reports that egg quality classification clusters are mainly grouped depending on their commercial size. In addition, the Leghorn’s egg groups differed from the rest of those from the Utrerana varieties, except for those of White Utrerana XL and Leghorn XL eggs, which reported a certain closeness. This may be indicative of the hybridization of the White Utrerana with the Leghorn breed, both with white plumage, which may have been historically performed by breeders as an attempt to decrease the consanguinity of the white variety, which is the Utrerana variety accounting with the smallest number of animals and the one which faces the highest endangerment risk.

Additionally, the present study may confirm the fact that product differentiation could be a feasible opportunity for the eggs of Utrerana varieties, which could constitute a favorable point when compared to eggs from other breeds that have traditionally been sold in the market [73].

The present discriminating tool allows to efficiently classify eggs based on quality-related traits as supported by the 73.2% of observations being correctly classified within their group. In this regard, weight traits play a pivotal role in the determination of the commercial quality of eggs.

All eggs belonging to the S category in White, Franciscan and Partridge, and L category of Franciscan and Partridge were correctly classified (Appendix A). However, 45.5% of M category Partridge eggs were classified as Franciscan M. Previous research suggested Partridge and Franciscan varieties present a significantly heavier yolk and slightly lower weight than the rest of the varieties or the Leghorn breed [14], hence similarities between egg quality-related traits of these two varieties could be expected. Moreover, 26.7% of XL category Leghorn eggs were classified as Leghorn L eggs, which may be explained as commercial genotypes have been selected to produce rather homogenous eggs, which may translate to a reduction in differences [74,75]. Furthermore, it may be worth mentioning that 23.8% of M category White Utrerana eggs were classified as M category Leghorn ones, with the likely hybridization between these two strains being the potential source for these similarities.

## 5. Conclusions

The present discriminating method has been proved and validated as an efficient tool to correctly classify eggs considering both external and internal traits. Additionally, this research confirms the fact that product differentiation could be a feasible opportunity for the eggs of Utrerana varieties, which could constitute a favorable point when compared to eggs from other breeds that have traditionally been sold in the market. Weight traits play a pivotal role in the determination of the commercial quality of eggs. This may evidence a patent lack of commercial attention provided to quality traits in favor of quantitative traits. However, this commercial strategy may be erroneous given it may not match the current general trend of customers preferring egg quality over quantity. Partridge and Franciscan classification confusion may derive from the fact that these varieties present significantly heavier yolks and slightly lower weights. Similarities between the eggs of White Utrerana and Leghorn hens may evidence reminiscences of hybridization.

## Figures and Tables

**Figure 1 foods-10-00632-f001:**
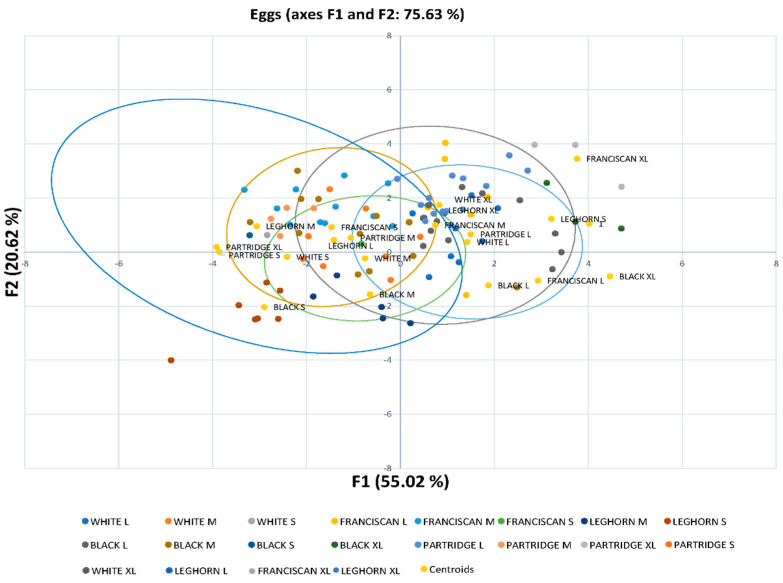
Territorial map depicting the eggs considered in the canonical discriminant analysis sorted across commercial quality categories (S, M, L, and XL) and laying hens genotypes (Leghorn and White, Black Franciscan, and Partridge Utrerana varieties). Centroids or canonical group means are the means for each group’s canonical observation scores. The larger the difference between centroids, the better the predictive power of the canonical discriminant function in classifying observations.

**Figure 2 foods-10-00632-f002:**
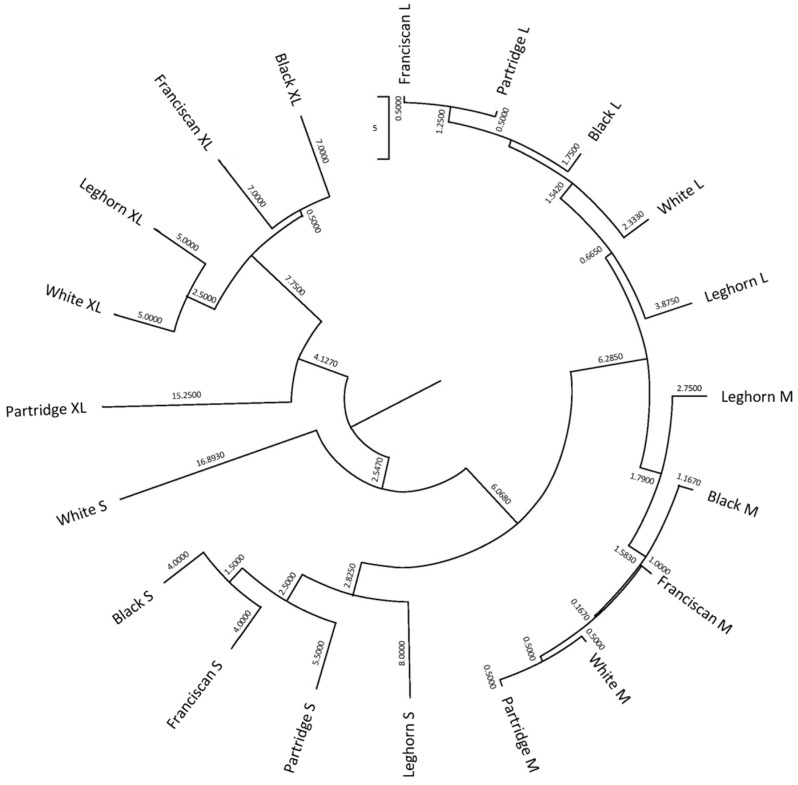
Cladogram constructed from Mahalanobis’s distances between commercial quality classification categories and laying hen genotypes.

**Table 1 foods-10-00632-t001:** Flock management information. All cages were chosen according to Council Directive 1999/74/EC of 19 July 1999, laying down minimum standards for the protection of laying hens.

Flock Management Parameter		Utrerana Varieties				Leghorn (Control)
		White	Franciscan	Black	Partridge	
Laying hens		15	15	15	15	10
Subgroups	Hens (70 weeks old)	8	8	8	8	0
	Pullets (28 weeks old)	7	7	7	7	10
Stocking density ^1^		4 animals per m^2^				
Nest box density ^1^		29 animals per m^2^				
Waterer allotment/space		Circle waterers of 5 cm in diameter per animal				
Feeder allotment/space		41 cm per animal				
Floor substrate		Wood shavings covering the cage floor at a depth of approximately 1 cm				
Nest box substrate		Plastic turf mats covering the floor at a depth of approximately 1 cm				

^1^ Stocking density and nest box density were determined after computing the whole cage’s surface considering each cage’s dimensions were 50 × 62 × 41 cm and its surface area was 0.25 m^2^.

**Table 2 foods-10-00632-t002:** Number of observations (eggs) classified per breed/variety and commercial size.

	S (<53 g)	M (53–63 g)	L (63–73 g)	XL (>73 g)	Total
White	2	46	45	3	96
Franciscan	7	71	25	2	105
Black	8	43	34	10	95
Partridge	8	32	32	3	75
Leghorn	12	83	60	15	170
Total	37	275	196	33	541

**Table 3 foods-10-00632-t003:** Multicollinearity analysis of quality-related traits of eggs to discard for redundant variables.

Statistics/Parameters	Tolerance (1 − R^2^)	VIF
Shell L*	0.2042	4.8980
Shell a*	0.2657	3.7642
Yolk weight	0.4428	2.2585
Yolk diameter	0.4504	2.2204
Eggshell strength	0.5250	1.9047
Eggshell weight	0.5526	1.8096
Area under the force–displacement curve	0.5733	1.7441
Yolk color	0.5877	1.7016
Yolk a*	0.6125	1.6326
Yolk b*	0.6184	1.6171
Albumen weight	0.6744	1.4829
Eggshell thickness	0.7037	1.4212
Yolk L*	0.7044	1.4196
Haugh units	0.7541	1.3261
Albumen pH	0.8314	1.2027
Yolk pH	0.8445	1.1841
Shape index	0.8735	1.1448

Interpretation thumb rule: variance inflation factor (VIF) = 1 (not correlated); 1 < VIF < 5 (moderately correlated); VIF ≥ 5 (highly correlated). VIFs > 5 are not presented in the table.

**Table 4 foods-10-00632-t004:** Summary of the results of Pillai’s trace of equality of covariance matrices of canonical discriminant functions to determine the idoneity of data for discriminant canonical analyses to be performed.

Parameter	Value
Pillai’s Trace Criterion	2.5016
F (Observed value)	4.7313
F (Critical value)	1.1357
df1	323
df2	8857
Significance	<0.0001
alpha	0.05

F, Snedecor’s F; df1, numerator degrees of freedom for the F-approximation; df2, denominator degrees of freedom for the F-approximation.

**Table 5 foods-10-00632-t005:** Canonical variable functions and percentages of self-explained and cumulative variance.

Function	Eigenvalue	Variance, %	Canonical Correlation	Cumulative Variance, %
F1	3.2788	55.0163	0.8754	55.0163
F2	1.2287	20.6172	0.7425	75.6335
F3	0.4021	6.7477	0.5355	82.3812
F4	0.3055	5.1253	0.4837	87.5064
F5	0.2160	3.6241	0.4215	91.1306
F6	0.1430	2.4002	0.3538	93.5307

**Table 6 foods-10-00632-t006:** Canonical Discriminant analysis efficiency parameters to determine the significance of each canonical discriminant function.

Test of Functions	Wilks’ Lambda	Chi Square	df	Significance
1 through 17	0.011	1320.792	323	<0.001
2 through 17	0.063	802.254	288	<0.001
3 through 17	0.155	541.386	255	<0.001
4 through 17	0.254	398.607	224	<0.001
5 through 17	0.358	298.246	195	<0.001
6 through 17	0.474	217.171	168	0.010

df: degrees of freedom.

**Table 7 foods-10-00632-t007:** Results for the tests of equality of group means to test for difference in the means across egg groups once redundant variables have been removed.

Variables	Rank	Wilks’ Lambda	F	df1	df2	Significance
Albumen weight	1	0.335	54.380	19	521	<0.0001
Eggshell weight	2	0.539	23.440	19	521	<0.0001
Yolk weight	3	0.566	21.010	19	521	<0.0001
Yolk diameter	4	0.651	14.680	19	521	<0.0001
Haugh units	5	0.700	11.760	19	521	<0.0001
Yolk b*	6	0.764	8.470	19	521	<0.0001
Shape index	7	0.800	6.860	19	521	<0.0001
Yolk color	8	0.812	6.360	19	521	<0.0001
Area under the force–displacement curve	9	0.832	5.550	19	521	<0.0001
Eggshell strength	10	0.837	5.350	19	521	<0.0001
Shell L*	11	0.844	5.080	19	521	<0.0001
Yolk a*	12	0.845	5.050	19	521	<0.0001
Shell a*	13	0.870	4.090	19	521	<0.0001
Eggshell thickness	14	0.892	3.320	19	521	<0.0001
Yolk L*	15	0.908	2.780	19	521	<0.0001
Yolk pH	16	0.941	1.710	19	521	0.0300
Albumen pH	17	0.957	1.250	19	521	0.2200

F, Snedecor’s F; df1, numerator degrees of freedom for the F-approximation (groups minus 1); df2, denominator degrees of freedom for the F-approximation (observations minus 1).

**Table 8 foods-10-00632-t008:** Discriminant loadings for external and internal quality–related traits determining the relative weight of each trait on each canonical discriminant function.

	F1	F2	F3	F4	F5	F6
Eggshell strength	−0.20	−0.26	−0.15	0.51	0.33	−0.06
Yolk L*	−0.08	0.02	0.07	−0.10	0.51	0.18
Yolk a*	−0.07	−0.35	−0.13	−0.53	−0.11	−0.03
Shape index	−0.04	−0.04	0.69	−0.13	−0.05	0.27
Shell a*	−0.04	0.05	0.09	0.64	−0.22	−0.62
Yolk pH	−0.03	0.13	−0.12	−0.07	0.11	0.32
Haugh units	−0.03	−0.45	0.00	0.21	0.37	−0.27
Yolk b*	−0.02	0.44	−0.01	0.15	−0.47	−0.48
Area under the force–displacement curve	0.01	0.28	0.43	0.02	−0.06	0.04
Albumen pH	0.02	0.07	0.07	0.03	−0.10	−0.04
Shell L*	0.04	−0.04	0.14	0.60	0.02	−0.55
Yolk color	0.05	0.31	0.37	−0.08	−0.01	0.28
Eggshell thickness	0.06	0.06	0.23	0.29	0.09	0.05
Yolk diameter	0.08	0.35	−0.12	−0.08	0.29	−0.36
Yolk weight	0.20	0.43	0.12	0.12	0.43	0.08
Eggshell weight	0.56	−0.20	0.40	−0.33	−0.07	−0.47
Albumen weight	0.83	−0.07	−0.21	0.30	−0.27	0.27

## Data Availability

Data will be made available from the corresponding author upon reasonable request.

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
