# Peer review of "Discriminant Canonical Analysis as a Validation Tool for Multivariety Native Breed Egg Commercial Quality Classification"

_foods, 2021, doi:10.3390/foods10030632_

Round 1

Reviewer 1 Report

First at all, I want to thank for the possibility to review this manuscript, which is interesting, overall well-written and contains interesting information for the scientific community. The methods used seems to be appropriate and correctly done.

Nevertheless, I have some issues which I would like to be clarified.

Abstract seems to be a bit too technical. The last two sentences can be omitted. Please try make it more reader-friendly, maybe by adding some sentences similar to those in conclusion section.

Varieties: white, franciscan, black and partridge - uppercase or lowercase - please unify throughout the manuscript.

Term “resistance area” is incorrect. In physics, area under the force-displacement curve is equal to work done, i.e. work needed to the crack the eggshell in this case. Please correct throughout the manuscript.

L67 Please remove “for instance”

L167 Delrin – please change to polyoxymethylene (POM) - delrin is the DuPont registered trademark

L166-168 Results of which measurement are presented?

L171 eggshell thickness- and average result is reported?

L184-188 Sentence unclear. Grammar?

L223 .... where R2 is the coefficient of determination of the regression equation. (all symbols in formulas must be explained)

L256 … (PCA).

L269 I am not a statistician and I could be wrong, but isn’t superfix T denoting matrix transpose missing?

L279 please add some information about this software (developer? homepage?)

L286 “obtained by chance”

L289 Please correct to “Press's Q= [N-(nK)}^2/[N(K-1)]” (superscript K and parenthesis for numerator)

L292 “…with one degree of freedom”

L299 “… VIF for other variables are shown..”

Table 3 footnote: please correct the figure footnote (VIFs >5 are not presented in the table) or add these removed variables to the table.

Table 4 title: lowercases

L313  Please consider changing to “Functions F1 and F2 with eigenvalues greater than 1 explains…”

L314 “..75.68% of total variance, while….”

Tables: Please consider unifying: Sig. or p-value. Tables 6 & 7: Some obvious yet necessary explanation are needed in tables caption (df, f1, df2...)

Table 7: Please move “rank column” to the beginning of the table

Figure 1. Figure is not easy to follow. Please consider changing it into a table with standardized coefficients for each variable for each discriminant function. It will allow the readers to calculate for themself resulting discriminant functions F1 and F2. (also figure contains commas instead of dots in y-axis)

Figure 2. Figure it is not clear enough. Please add figure legend (what do the colors mean?)

L345 correct to “group”

Figure 3. Nei's distance ? The information how to calculate Nei's distance form Mahalanobis distance is missing.

L355-356 73.2% and 57.1% (dots)

L357 please give obtained N, n and K values used for Press's Q calculation

L383- Which results for shell a* and shell *b? I do not understand.

L485. Table S1 title: "Appropriately classified eggs (%) according..."

L556-557: Ref [25] As this is still ahead of print, please add doi number (doi.org/10.2478/aoas-2020-0112)

Author Response

Reviewer 1

Comments and Suggestions for Authors

First at all, I want to thank for the possibility to review this manuscript, which is interesting, overall well-written and contains interesting information for the scientific community. The methods used seems to be appropriate and correctly done.

Response: We thank the reviewer for his/her kind comments.

Nevertheless, I have some issues which I would like to be clarified.

Abstract seems to be a bit too technical. The last two sentences can be omitted. Please try make it more reader-friendly, maybe by adding some sentences similar to those in conclusion section.

Response: The abstract was revised and rewritten to make it more reader friendly, but still leaving the main statistial results.

Varieties: white, franciscan, black and partridge - uppercase or lowercase - please unify throughout the manuscript.

Response: Manuscript was checked and all the names of varieties were capitalized.

Term “resistance area” is incorrect. In physics, area under the force-displacement curve is equal to work done, i.e. work needed to the crack the eggshell in this case. Please correct throughout the manuscript.

Response: The term Resistance Area was changed to Area under the force-displacement curve as suggested by the reviewer.

L67 Please remove “for instance”

Response: We removed “For instance.”

L167 Delrin – please change to polyoxymethylene (POM) - delrin is the DuPont registered trademark

Response: We changed delrin to polyoxymethylene (POM)as suggested by reviewer.

L166-168 Results of which measurement are presented?

Response: Both results were presented. We clarified it in the body text. Spectrophotometer was considered through L*, a* and b* mesurements while Roche fan was represented by Yolk Color trait.

L171 eggshell thickness- and average result is reported?

Response: Yes, eggshell thickness typically includes the membrane using a modified micrometer and averaging three measurements around the blow-hole near the equator of the egg as suggested by Klaas et al. 1974. We clarified it in text.

Klaas, E. E., Ohlendore, H. M., & Heath, R. G. (1974). Avian eggshell thickness: variability and sampling. The Wilson Bulletin, 156-164.

L184-188 Sentence unclear. Grammar?

Response: We agree with the reviewer. The sentence was rewritten.

L223 .... where R2 is the coefficient of determination of the regression equation. (all symbols in formulas must be explained)

Response: Added.

L256 … (PCA).

Response: Added.

L269 I am not a statistician and I could be wrong, but isn’t superfix T denoting matrix transpose missing?

Response:

Mahalanobis distance can be defined as the reviewer suggested using the following formula, which includes the transpose, d(x,y)=(x−y)T Σ−1(x−yd(x,y)=(x−y)TΣ−1(x−y), where Σ is an estimate of the covariance matrix for some data; this implies it is symmetric. If the columns used to estimate Σ are not linearly dependent, Σ is positive definite. Symmetric matrices are diagonalizable and their eigenvalues and eigenvectors are real. Symmetric matrices have eigenvalues which are all positive. The eigenvectors can be chosen to have unit length, and are orthogonal. One of the properties of orthogonal matrices is that their inverse equals their transpose.

Conclusively, there are different formulas for Squared Mahalanobis distances. The one used here was that reported by Kurnianto, et al. (2013) as follows;

“The between-breed squared Mahalanobis distance matrix was computed as: Mahalanobis distance that is written as: Dij 2 = (Xi-Xj)' Cov-1 (Xi-Xj) Where: Dij 2 : distance between ith breed and jth breed. Cov-1: the inverse of the covariance matrix of measured variable X. Xi and Xj: are the means of variable X in ith breed and jth breed, respectively.”   

Kurnianto, E., Sutopo, S., Purbowati, E., Setiatin, E. T., Samsudewa, D., & Permatasari, T. (2013). Multivariate analysis of morphological traits of local goats in Central Java, Indonesia. Iranian Journal of Applied Animal Science, 3(2), 361-367.

L279 please add some information about this software (developer? homepage?)

Response: Added.

L286 “obtained by chance”

Response: Changed.

L289 Please correct to “Press's Q= [N-(nK)}^2/[N(K-1)]” (superscript K and parenthesis for numerator)

Response: Corrected.

L292 “…with one degree of freedom”

Response: Changed.

L299 “… VIF for other variables are shown..”

Response: Changed.

Table 3 footnote: please correct the figure footnote (VIFs >5 are not presented in the table) or add these removed variables to the table.

Response: Changed.

Table 4 title: lowercases

Response: Changed.

L313  Please consider changing to “Functions F1 and F2 with eigenvalues greater than 1 explains…”

Response: Changed.

L314 “..75.68% of total variance, while….”

Response: Added.

Tables: Please consider unifying: Sig. or p-value. Tables 6 & 7: Some obvious yet necessary explanation are needed in tables caption (df, f1, df2...)

Response: Information added.

Table 7: Please move “rank column” to the beginning of the table

Response: Rank column was moved to the beginning.

Figure 1. Figure is not easy to follow. Please consider changing it into a table with standardized coefficients for each variable for each discriminant function. It will allow the readers to calculate for themself resulting discriminant functions F1 and F2. (also figure contains commas instead of dots in y-axis)

Response: Figure 1 was changed to Table 8.

Figure 2. Figure it is not clear enough. Please add figure legend (what do the colors mean?)

Response: Figure 2 legend was included.

L345 correct to “group”

Response: Changed.

Figure 3. Nei's distance ? The information how to calculate Nei's distance form Mahalanobis distance is missing.

Response: This was a mistake. We corrected it, we only used Mahalanobis distances.

L355-356 73.2% and 57.1% (dots)

Response: Corrected.

L357 please give obtained N, n and K values used for Press's Q calculation

Response: Values required were added.

L383- Which results for shell a* and shell *b? I do not understand.

Response: We changed results to coordinates. We meant the values of these parameters

L485. Table S1 title: "Appropriately classified eggs (%) according..."

Response: Added.

L556-557: Ref [25] As this is still ahead of print, please add doi number (doi.org/10.2478/aoas-2020-0112).

Response: Added.

Reviewer 2 Report

The work is very interesting: it is concerning poultry biodiversity and the use of a statistical tool as Canonical Discriminant Analysis.

The description concerning the statistical part is in a very technical form , thus some more clear descriptions of the data shown in tables and figures should be added, when and where it is possible, for providing a more comprehensible paper to the readers of Foods journal.

The text shows some typing mistakes, check it.

Row 104: individual cages were used. In Table 1 stocking density, nest box density, floor substrate shows not clear data.

Table 1 : the age of the hens was not the same for all the strain: the age of the hen is an important factor affecting the quality of the egg, and this factor should be taken into account for this study.

Row 117: a complete laying cycle of 1 year for the study. 541 eggs studied. Which was the interval of sampling?

Row 119-120: which is the meaning of this sentence?

Row 147: I suggest to change the subtitle(measurements on eggs..)

Row 149: the quality was registered daily for the whole duration of the study: why a low number of collected eggs?

Row 161: eggshell strenght and area: did you consider these traits for DCA?

Row 181-183: I suggest two sentences instead of one very long.

Row 184-185: it is not clear.

For the statistical description: significance or p value, 0.05.

For the results, I suggest to add to the text a better description of the data shown in each figure and table.

Table 6: significance .

Row 318-322: two sentences, instead of one.

Table 7: why 19 for df1? I suggest to add short description for each table or figure.

Row 354-359: comma for decimals!!

Row 382-387; 390-392: it is not clear.

Row 404-405: why energy reserves in the albumen?

Row 406-407: lower than?

Row 412-423; 426-430: more details, it is not clear.

Row439-443; 478-482: it is not clear.

Author Response

Reviewer 2

Comments and Suggestions for Authors

The work is very interesting: it is concerning poultry biodiversity and the use of a statistical tool as Canonical Discriminant Analysis.

Response: We thank the reviewer for his/her kind comments.

The description concerning the statistical part is in a very technical form , thus some more clear descriptions of the data shown in tables and figures should be added, when and where it is possible, for providing a more comprehensible paper to the readers of Foods journal.

Response: We agree with the reviewer. Suggestion was followed and information was clarified and expressed in a rather less technical manner.

The text shows some typing mistakes, check it.

Response: The whole manuscript was revised and checked for typos and grammar mistakes and to improve readability.

Row 104: individual cages were used. In Table 1 stocking density, nest box density, floor substrate shows not clear data.

Response: Information was clarified.

Table 1 : the age of the hens was not the same for all the strain: the age of the hen is an important factor affecting the quality of the egg, and this factor should be taken into account for this study.

Response: The selection of the individuals in the sample was performed considering both breeds commercial cycle and the age when hens of both breeds reach 50% of laying (egg production during a laying cycle). Contextually, the typical production cycle in commercial layers (Leghorn hens among others) lasts about 72 weeks [44]. However, this cycle may extend until 156 weeks in around a third of the Utrerana population [45]. Additionally, according to Kuo, et al. [46], the age at sexual maturity is estimated by age in weeks when 50% egg production is reached. In these regards, the same authors suggested the age when 50% egg production in White Leghorn is reached around 21 weeks. By contrast, the information reported by Orozco Piñán [45] suggested the average age of Utrerana hens at the moment of the first laying was 25 weeks. Furthermore, the breeding criterion of both breeds may defer as while White Leghorn hens breeders have traditionally selected animals for precocity [47], Utrerana breeders have not sought this trait as a priority rather benefiting from the natural lay cycle of the breed [45]. Zita, et al. [48] suggested egg quality characteristics are affected by the interaction of genotype (breed and strain) and hen’s age, rather than exerting their effects independently.

Row 117: a complete laying cycle of 1 year for the study. 541 eggs studied. Which was the interval of sampling?

Response: We have noticed there was a mistake. Quality traits were registered fortnightly.

Row 119-120: which is the meaning of this sentence?

Response: We clarified the sentence in the body text.

Row 147: I suggest to change the subtitle(measurements on eggs..)

Response: Changed.

Row 149: the quality was registered daily for the whole duration of the study: why a low number of collected eggs?

Response: We have noticed there was a mistake. Quality traits were registered fortnightly.

Row 161: eggshell strenght and area: did you consider these traits for DCA?

Response: Yes, we clarified this and eggshell area was renamed to area under the force-displacement curve.

Row 181-183: I suggest two sentences instead of one very long.

Response: We followed the reviewer’s suggestion.

Row 184-185: it is not clear.

Response: We agree with the reviewer. The sentence was rewritten.

For the statistical description: significance or p value, 0.05.

Response: Changed.

For the results, I suggest to add to the text a better description of the data shown in each figure and table.

Response: Suggestion was followed.

Table 6: significance .

Response: Following the suggestion by another reviewer Sig. Were unifed as p-values.

Row 318-322: two sentences, instead of one.

Response: Suggestion was followed.

Table 7: why 19 for df1? I suggest to add short description for each table or figure.

Response: Short description was added.

Row 354-359: comma for decimals!!

Response: Corrected.

Row 382-387; 390-392: it is not clear.

Response: We clarified it in the body text.

Row 404-405: why energy reserves in the albumen?

Response: We clarified it in the body text.

Row 406-407: lower than?

Response: Yes, it was clarified in the body text.

Row 412-423; 426-430: more details, it is not clear.

Response: We clarified the information and added further information.

Row439-443; 478-482: it is not clear.

Response: We clarified it in the body text.

Round 2

Reviewer 2 Report

The paper has been improved. I indicate other little suggestions:

row 29: 28.7%

row 67: delete for instance

row 69-70: ..dependent mainly on laying hen's age and nutritional factors

row 72: delete for instance

table 1: hens and pullets should be indicated as sub-groups

row 240-244: check it, maybe a typing error

table 6:the  significance level should be indicated better

row 406: hybrid commercial cycle, I suppose! and both genotypes reach... 

row 448: one of the source..

row 462: energy or protein?

Check the text for typing mistakes.

Author Response

Reviewer 2

The paper has been improved. I indicate other little suggestions:

Response: We thank the reviewer for his/her kind comments.

row 29: 28.7%

Response: Changed.

row 67: delete for instance

Response: Deleted.

row 69-70: ..dependent mainly on laying hen's age and nutritional factors

Response: Changed.

row 72: delete for instance

Response: Deleted.

table 1: hens and pullets should be indicated as sub-groups

Response: Changed.

row 240-244: check it, maybe a typing error

Response: Corrected.

table 6:the  significance level should be indicated better

Response: We agree. We followed the examples in other papers published in Foods this time.

row 406: hybrid commercial cycle, I suppose! and both genotypes reach... 

Response: We agree. Changed.

row 448: one of the source..

Response: Changed.

row 462: energy or protein?

Response: Indeed both, but here we referred to energy as it was discussed in the paragraph below.

Check the text for typing mistakes.

 Response: Text was checked for typos and grammar mistakes.